# Standardized approach to the conservative surgery of hepatic cystic echinococcosis: A prospective study

**Aymen Trigui[1], Sami Fendri[1], Mohammad Saad Saumtally** **[2]\*, Amira Akrout[1], Jihen Trabelsi[2], Rahma Daoud[1], Nozha Toumi[3], Salma Ketata[4], Wael Boujelbene[1], Rafik Mzali[1], Chadli Dziri[5], Mohamed Ben Amar[1], Salah Boujelben[1]**

**1** University of Sfax, Faculty of Medicine; Department of General and Digestive surgery. Habib Bourguiba Hospital, Sfax, Tunisia, **2** University of Sfax, Faculty of Medicine; Department of Epidemiology. Hédi Chaker Hospital, Sfax, Tunisia, **3** University of Sfax, Faculty of Medicine; Department of Radiology. Habib Bourguiba Hospital, Sfax, Tunisia, **4** University of Sfax, Faculty of Medicine; Department of Anaesthesiology. Habib Bourguiba Hospital, Sfax, Tunisia, **5** University of Tunis, General Surgery; Honoris Medical Simulation Centre director, Tunisia

\* saad0991@gmail.com

## Abstract

### Objective

Surgery is the mainstay of hepatic cystic echinococcosis (HCE). The conservative surgery of HCE carries a non-negligible risk of recurrence and significant morbidity, dominated by Deep Surgical Site Infections (DSSI). To address these issues, we have improved and standardized this technique, which could reduce complications and achieve better postoperative outcomes.

### Patients and methods

We conducted a prospective study from June 2017 to June 2022 involving of patient operated using a standardized open technique for uncomplicated HCE at Habib Bourguiba University Hospital, Sfax, Tunisia. The aim was to obtain results at least similar to radical management in terms of DSSI. Patients with large cystobiliary fistulas or patients with complicated cysts were excluded.

### Results

Fifty patients with 106 cysts were operated using the standardized technique comprising of liver mobilization, intraoperative ultrasound, systematic methylene blue injection to detect cystobiliary fistulas and omentoplasty. The median age of the patients was 44(semi-interquartile range: 16) years. The main symptom described by the patient was pain in 43 cases (86%). An abnormal liver test was found in 20 cases (40%). On imaging studies, the cyst had a median size of 7.4(3.0) cm. Cyst of the hepatic dome accounted for 38 cases (35.8%) with most cysts being situated in the right hemi-liver. Visual inspection of the cavity and Methylene blue testing allowed for the discovery of 57 cysts (53.7%) that had cystobiliary fistulas that were sutured. Omentoplasty was performed in 77 cysts (72.6%). Postoperatively,

**Data Availability Statement:** The data are not publicly available due to their containing information that could compromise the privacy of research participants. For access to the data,

please contact the local ethics committee(Comité de Protection des Personnes adapté à l'Expérimentation Médicale ou Scientifique des Produits Médicaux destinés à la Médecine Humaine) at Rue Firdaous, 3029 Sfax. Email: cppsud.tunisie@gmail.com.

**Funding:** The author(s) received no specific funding for this work.

**Competing interests:** The authors have declared that no competing interests exist.

only 2 cases (1.9%) developed a DSSI in the form of an external bile leak with resolved with conservative management. No case of recurrence was found after a median follow-up of 24 months.

## Conclusion

The standardized conservative surgical technique, in selected patients, shows promise in reducing DSSI rates and overall morbidity, and achieve as equally good result as radical management.

## Author summary

This study addresses the treatment of hepatic cystic echinococcosis (HCE), a serious health concern in many parts of the world. Surgery has been the mainstay of the management which could be conservative or radical. Conservative surgery has high recurrence rates and complications as opposed to radical surgery. Through a prospective study, we showed that an improved and standardized a surgical technique of the conservative approach could achieve postoperative outcomes as good as the radical management. The standardized approach significantly reduced Deep Surgical Site Infections (DSSI) as well as overall morbidity rates in patients. This was achieved through various adjuncts such as methylene blue test, liver detachment, and omentoplasty.

## Introduction

Cystic echinococcosis (CE) is a zoonotic disease that is endemic in certain parts of Eurasia, Africa, Australia, and South America [1] with poses a significant public health problem. Currently, the management of Hepatic Cystic Echinococcosis (HCE) involves various available therapeutic options such as medical, percutaneous, endoscopic, and surgical approach. The goal of treatment is to eradicate the parasite while avoiding complications related to the residual cavity. There are two main surgical approaches to treat HCE: conservative treatment and radical treatment [2].

The most common conservative surgical technique in endemic areas is partial cystectomy (PC) as described by Lagrot in 1957 [3,4]. This technique is simple and quick to perform, requires few resources, and has a low risk of major bleeding. As opposed to the latter technique, radical management which entails liver parenchymal resections with a non-negligible bleeding risk [2,5].

Some of the studies conducted, advocates for radical management citing less operative morbidity and less recurrence [5,6]. Despite these promising results, these studies had certain limitations such as exclusion of complicated cyst cases or multiple cysts and varied in their criteria for selecting patients for radical surgery.

Moreover, the radical technique is more complex and requires a certain level of expertise and/or may be not feasible in cases of liver resections with proximity to major vessels or in cases of multicystic disease. In such cases, PC remains a viable option. The two techniques are not mutually exclusive but rather complementary. Therefore, the idea of improving the PC through standardization arose in order to reduce the significant morbidity associated with this technique. Several technical refinements, such as liver mobilization, intraoperative ultrasound

[7], systematic methylene blue injection to detect cystobiliary fistulas [8] and omentoplasty [9] have been added to the traditional technique to improve postoperative outcomes.

In this study, we present the results of a prospective study of patients undergoing surgery for HCE using the standardized technique.

The primary objective of this study was to compare the results of the standardized technique in terms of specific postoperative morbidity, in particular Deep Surgical Site Infections (DSSI), as compared to radical management.

## Patients and methods

### Ethics statement

This study was conducted following the Helsinki Declaration. Informed written consent was obtained from each subject. Prior to the start of our study, approval was obtained from the local ethics committee (Comité de Protection des Personnes adapté à l'Expérimentation Médicale ou Scientifique des Produits Médicaux destinés à la Médecine Humaine), under reference "CPP sud numéro 0021/2017." This committee is the sole medical or scientific research centre for human medicine in our institution, the University of Sfax. It is approved by the Ministry of Public Health. All patients who participated in our prospective study signed a written consent form.

### Study design

Our study focuses on patients who underwent conservative surgery (PC) for HCE at the Department of General Surgery of the Habib Bourguiba University Hospital in Sfax. The prospective cohort, consists of patients who underwent surgery using the standardized technique from June 2017 to June 2022. This trial was registered with PACTR.org: PACTR202312729770861.

We included in our study all patients who were operated for HCE in an emergent or elective setting by laparotomy. We excluded patients who had cysts complicated with a large cystobiliary fistula revealed by acute cholangitis or acute pancreatitis or presence of cyst debris in the common bile duct on preoperative imaging which could manifest itself through a common bile duct dilation or direct visualization of obstacle in the common bile duct.

We distinguish 2 types of cysts: cysts with overt, large CBF and cysts presenting occult fistulas. Management differs for these 2 types. For large and peripheral cysts, radical resection is preferred. However, cysts with close proximity to major veins and biliary structures, internal trans-fistulary drainage or cystobiliary disconnection (Perdromo technique) is performed since simple suturing of the fistula within the pericyst is not effective. Thus, only patients with occult fistulas were included, to get a homogeneous population and minimize selection bias.

### Surgical procedure for the standardized technique

All patients were operated by laparotomy by 2 designated surgeons. We opted for a right subcostal incision for anterior single cysts, sometimes extended to the left for HCE of the left lobe, or a Makuuchi incision in case of multiple hepatic hydatid lesions and/or posterior HCE. Liver mobilization is performed by cutting the round ligament and the falciform ligament 1 cm from the surface of the liver. Mobilization of the liver through sectioning of the right or left triangular ligament is only performed to expose a cyst located in the posterior superior region, particularly on the dome or segment II. These manoeuvres allow better exposure to the cyst and may allow easier inspection of the cyst wall. The extent of this mobilization is adjusted

according to the location and relationships of the HCE. Visual and manual exploration of the cysts is carried out through inspection and palpation.

Intraoperative ultrasound is performed using a convex probe with a frequency range of 3 to 5 MHz. The pericyst is carefully examined, and vascular and biliary relationships, wall thickness, presence of calcifications, and the presence of exocysts are noted which are the primary source of recurrence [7].

The peritoneal cavity is protected by using laparotomy sponges soaked in 20% hypertonic saline solution. In case of CE1 and CE3a detected on intraoperative ultrasound, the cyst is carefully punctured with a 15G needle and the liquid is initially, slowly aspirated using a number of 50 cc syringes using a 3-way valve, Deflating the cyst helps to decrease the intracystic pressure and prevent accidental spillage when incising the cyst. In case of mastic contents (CE4 or CE3b), a 10 mm trocar without the mandrel is carefully inserted in the cyst wall and it is used to suck the mastic and lumpy contents which often cause blockages in suction systems. For multivesicular cysts such as in CE2 cysts, cyst aspiration is often unsuccessful because of the multiple daughter cysts. Therefore, the 10 mm trocar is used to aspirate cyst contents.

Once the intracystic pressure decreases and cyst becomes flaccid, a 2 cm incision is carefully performed on the pericyst. The cyst content is fully aspirated, and the cyst cavity is generously irrigated with normal saline, aspirating and replenishing the solution 2 to 3 times to thoroughly dilute any remaining protoscoleces. The cavity is then swabbed with gauze dipped in 20% hypertonic saline solution and it the swab is left in the cavity for 10 minutes. Direct washing with hypertonic saline solution is not done to prevent sclerosing cholangitis.

The cyst incision is widened to better asses the cavity and a visual exploration of the residual cavity is carried out to detect any bile leakage or visible cystobiliary fistula. The inner wall is gently scraped using a curette to remove membranous debris. Any visible CBFs detected during the visual exploration of the cavity are sutured using Vicryl 2/0.

A systematic cholecystectomy is performed, and a transcystic catheter is inserted for the Methylene Blue Test (MBT) to detect occult cysto-biliary fistulas. The dye is administered at low pressure and any leakage of the dye in the cyst cavity would be considered as cystobiliary fistulas (CBF) [8]. These are then sutured using Vicryl 2/0. Then, clamping the main bile duct during dye injection enables the identification of additional CBF and serves as a leakage test for the previously sutured fistula. The sutures that showed leakage of methylene blue are reinforced, and as well as any additional occult cystobiliary fistulas detected during this methylene blue test. The integrity of the sutures is checked by performing a methylene blue test with and without manual clamping of the common bile duct.

After the leakage tests are negative or in the absence of any CBFs, the cavity is thoroughly washed with hypertonic saline. The protruding part of the pericyst is then resected, preventing any live protoscoleces from leaking outside the cavity.

In case of a non-yielding and rigid pericyst intralamellar pericystectomy is performed to give a supple cavity wall. The greater omentum is partly detached from the transverse colon and the greater omentum is affixed to the residual cavity for omentoplasty, if possible, using a 2/0 Vicryl suture [9]. A passive drainage is inserted in the residual cavity through a stab incision in the right flank in all cases.

## Patient follow-up

As part of the protocol of study, food was allowed on post-operative day 1 with early mobilization and administration of oral albendazole, 400 mg twice daily. The drain was usually removed on post-operative day 5 before discharge unless a complication is detected.

After discharge from the hospital, the patients are followed up in the outpatient clinic of the surgical department. The latter takes care of perioperative albendazole treatment and follow-up for relapses. To detect recurrences early–depending on the location of the cyst–ultrasound or CT or MRI investigations are regularly performed.

### Definition of outcomes of the study

The main complication of conservative surgery of HCE which are:

- External Biliary Fistula: We considered any clear bile drainage from the abdominal drain, regardless of the timing, duration, or quantity, as a postoperative biliary fistula [10].

- Cavity infection: Defined as the occurrence of an infection in the residual cavity with or without the presence of drainage.

These aforementioned complications were grouped under the term "Deep Surgical Site Infection" (DSSI) [11]. Cavity infections result from bile accumulation by means of a CBF. External bile leakage shares the same cause. Therefore, the rationale behind grouping these complications together is that they both stem from the common cause. The primary outcome of our study was the occurrence of DSSI in the population which was main specific complication of the conservative surgery. Postoperative morbidity was assessed using the Clavien-Dindo classification [12].

The secondary outcomes were the overall morbidity, mortality and disease recurrence. Morbidity was defined as the occurrence of one or more complications during hospitalization or within the 30 days following the surgery and mortality was defined as is defined as death occurring within 30 days following the surgery or during the same hospitalization, regardless of its duration. Recurrent hepatic hydatid disease refers to the emergence of new active cysts following surgery.

### Statistical study

IBM SPSS Statistics for Windows, version 26 (IBM Corp., Armonk, N.Y., USA) was used for data entry and for all statistical analyses. Frequencies were calculated for qualitative variables, while the median (semi-interquartile range (SIR)) was computed for quantitative variables. For overall morbidity and nonspecific morbidity, the analysis was done per patient. For specific complications and cyst recurrence, the unit of study was the cyst. Therefore, for patients with two or more cysts, each cyst was considered independent.

We hypothesized that the standardized technique should achieve results which are at least good as the radical management. The sample size of 99 cysts for a single arm was calculated to give 95% power for detecting a statistically significant difference in terms of DSSI (with alpha-value 0.05), based on a previous meta-analysis [13] comparing DSSI rates in conservative surgery (14.6%) versus radical surgery (5.1%). We hoped to achieve DSSI rates using the standardised technique at least similar to the radical treatment in this single arm trial.

### Results

A flowchart of the eligibility criteria is presented in Fig 1. Following the exclusion of 18 cysts, 106 cysts were included in our study (50 patients) who underwent surgery for one or more cystic echinococcosis of the liver between July 2013 and June 2022.

The Table 1 and Table 2 resumes the data of the participants in terms of demographic parameters, clinical examination data, laboratory findings, and imaging data.

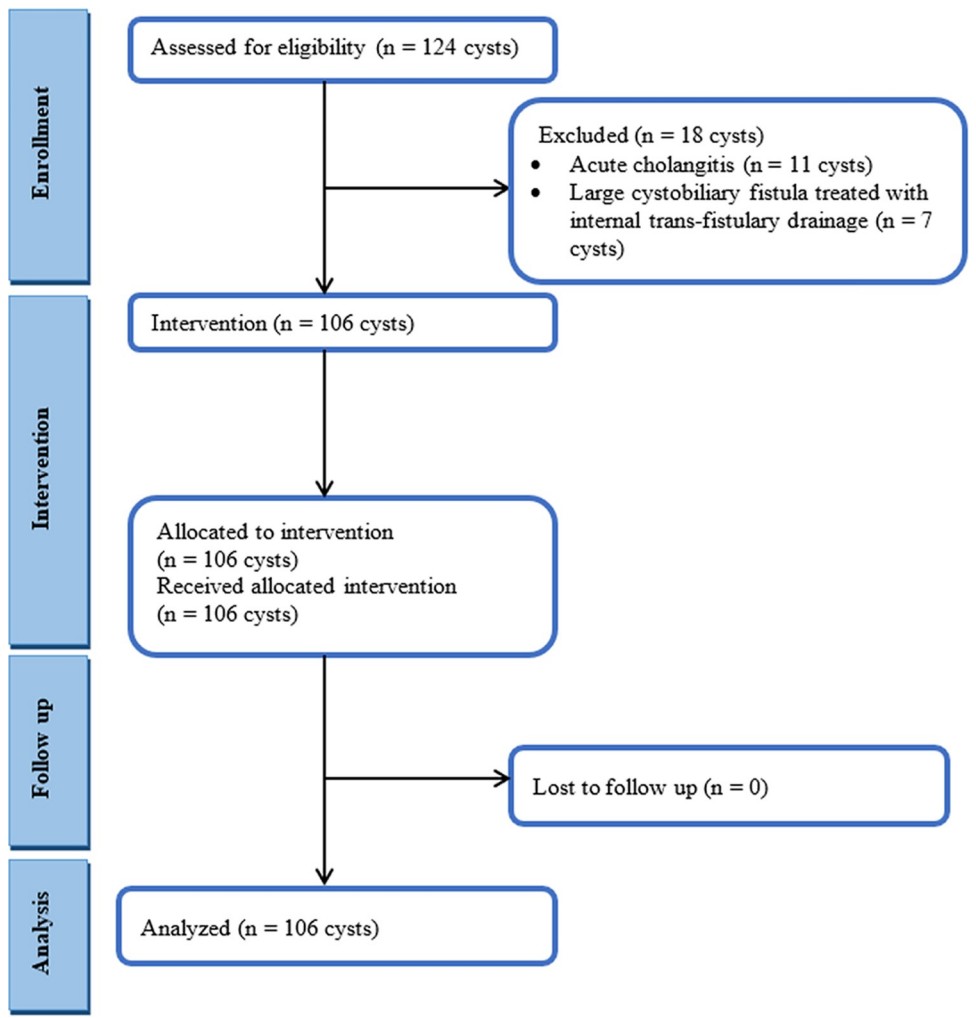

**Fig 1. Flow diagram of the inclusion and exclusion criteria.**

## Operative findings

Operative findings showed that most cysts were exophytic with a protruding dome and with clear cyst contents as showed by the Table 3.

In the study population, the Makuuchi J incision was the most commonly performed accounting for 60.0%. followed by the right subcostal incision in 14 cases (28.0%). The rest of the participants had bilateral subcostal incision and median incision depending in the cyst localisation, 39 patients (78.0%) underwent liver mobilization to improve exposure of posterior cysts.

## Occult cystobiliary fistulas

The MBT was performed in all patients in the population.

A total of 122 cystobiliary fistulas (CBF) were identified and sutured. Among the 48 cysts that were complicated by occult FKB (i.e. not evident on initial examination). the preoperative injection of methylene blue (MBT) during surgery allowed the detection of 106 additional CBF. Moreover, clamping of the common bile duct was performed, leading to the

**Table 1. Preoperative parameters of the patients in both groups.**

| N = 50 patients | | N (%) |
|---|---|---|
| **Characteristics** | | |
| **Age (median, SIR)** | | 44 (16) |
| **Gender** | Male | 17 (34.0) |
| | Female | 33 (66.0) |
| **Origin** | rural | 44 (88.0) |
| | urban | 6 (12.0) |
| **Recurrence of HCE at admission** | | 4 (8.0) |
| **Recurrence of PCE at admission** | | 4 (8.0) |
| **ASA Score** | ASA 1 | 42 (84.0) |
| | ASA 2 | 5 (10.0) |
| | ASA 3 | 3 (6.0) |
| **Physical examination** | | |
| **Incidental discovery** | | 7 (4.8) |
| **Abdominal pain** | | 43 (86.0) |
| **Nausea and emesis** | | 15 (30.0) |
| **Jaundice** | | 1 (2.0) |
| **Palpable abdominal mass** | | 5 (10.0) |
| **Abdominal tenderness** | | 28 (56.0) |
| **Hepatomegaly** | | 4 (8.0) |
| **Laboratory findings** | | |
| **Leucocytosis** | | 11 (22.0) |
| **Hypereosinophilia** | | 15 (31.3) |
| **CRP** | > 6 mg/l | 18 (38.3) |
| **Elevated ALT or AST** | | 6 (12.5) |
| **Cholestasis** | | 18 (40.9) |
| **Hyperbilirubinemia, Total bilirubin >25µmol/l** | | 2 (4.0) |
| **Abnormal Liver Function Test** | | 20 (40.0) |
| **Hemostasis tests** | Normal | 44 (88.0) |
| **Serologic test** | positive | 37 (86.0) |

SIR: semi-interquartile range, HCE: Hepatic Cystic Echinococcosis, PCE: Pulmonary Cystic Echinococcosis, CRP: C Reactive Protein, AST: aspartate aminotransferase, ALT: alanine aminotransferase

identification of 16 more CBF in 15 cysts. The median diameter of the fistula was 1.1 mm. We also found that cysts CBFs were larger (mean diameters of 6.1± 0.6 cm without CBFs vs 8.0 ± 0.5 cm with CBFs), p = 0.024.

At the end of these explorations, a total of 57 cysts complicated by cystobiliary fistulas (CBF) were discovered, accounting for 53.7% of cases.

## Management of the residual cavity

Omentoplasty was performed almost systematically whenever possible. A total of 77 cysts (72.6%) and capitonnage was performed in only 2 cysts (1.9%). A passive drainage was inserted in 101 cysts (95.3%).

## Analysis of the primary end-point: DSSI

We obtained a DSSI rate of 1.9% in our cohort which is at least better than DSSI rates of 5.1% as defined in our aims. These were 2 cysts that developed an external bile fistula in the

**Table 2. Preoperative characteristics the cysts in both groups.**

| N = 106 cysts | | N (%) |
|---|---|---|
| **Imaging findings** | | |
| **Number of cysts** | Solitary | 28 (54.9) |
| | 2 cysts | 11 (21.6) |
| | Multiple cysts | 12 (23.5) |
| **Cyst size in cm** | (median, SIR) | 7.4 (3.0) |
| **Calcifications** | | 14 (13.2) |
| **Visible laminated layer** | | 25 (23.6) |
| **Heterogeneous mass** | | 34 (32.1) |
| **Daughter cysts** | | 40 (37.7) |
| **Exocysts** | | 15 (14.2) |
| **WHO-IWGE classification** | CE 1 | 25 (23.6) |
| | CE 2 | 21 (19.8) |
| | CE 3a | 22 (20.8) |
| | CE 3b | 21 (19.8) |
| | CE 4 | 18 (17.0) |
| **Cyst localisation** | segment I | 1 (0.9) |
| | segment II | 19 (17.9) |
| | segment III | 20 (18.9) |
| | segment IV | 18 (17.0) |
| | segment V | 14 (13.2) |
| | segment VI | 17 (16.0) |
| | segment VII | 23 (21.7) |
| | segment VIII | 19 (17.9) |
| | Left hemiliver | 31 (29.2) |
| | Right hemiliver | 78 (73.6) |
| | Hepatic dome | 38 (35.8) |
| **Suspected complicated cyst** | | 57 (53.8) |
| **Biliary compression** | | 22 (20.8) |
| **Vascular compression** | | 46 (43.4) |

SIR: semi-interquartile range, WHO-IWGE: World Health Organization—Informal Working Groups on Echinococcosis

**Table 3. Intraoperative characteristics of the cysts in both groups.**

| | N (%) |
|---|---|
| **Type of cyst** | |
| Exophytic cyst | 101 (95.3) |
| Intrahepatic cyst | 5 (4.7) |
| **Characteristics of the pericyst** | |
| Supple pericyst | 82 (77.4) |
| Hard pericyst | 24 (22.6) |
| **Cyst contents** | |
| Clear | 94 (88.7) |
| Bile stained | 10 (9.4) |
| Purulent | 4 (3.8) |
| Gelatinous | 34 (32.1) |

postoperative period. In response to the fistula occurrence, intravenous antibiotic therapy with cefotaxime and metronidazole was initiated. No persistent fistulas beyond 30 days were found, and none of these patients required sphincterotomy or additional surgery. These bile leak appears on post-operative day 2 in both cases. The patients were started on antibiotic course of intravenous cefotaxime and metronidazole upon detection of the fistula. The average output of the fistula was 106 ml/day and it stopped being productive after a mean duration of 25 days. No need for sphincterotomy or surgery was required for these patients. Both patients had a hospital stay of 14 days.

No patients developed injury to bile ducts as a result of hypertonic saline use during the sterilization of the cyst.

## Secondary outcomes

We report an overall morbidity of 10% in the 50 patients studied with 3 patients having medical complications and 2 patients having the abovementioned DSSI. The median length of stay was 5.0 days with extremes of 1 day to 14 days.

The median follow-up duration for the patients was 24 months and no recurrence were detected on successive exams and imaging studies.

## Discussion

The standardized technique as described above demonstrates effective surgical management of cystic echinococcosis of the liver with low rates of complications, particularly DSSI (1.9%) which is at least as good as radical surgery. Moreover, our study also found an overall morbidity rate of 10% and no recurrence in the follow-up period. This could be attributed to the new improved techniques with adjuncts such as methylene blue test to detect occult CBF.

According to literature, several cohort studies a controlled randomized trial, and 3 meta-analyses [13–15] comparing the two mentioned techniques were found. These authors concluded that radical treatment is superior to conservative surgery in terms of Surgical Site Infections, overall morbidity, bile duct fistulas, and recurrence.

However, it should be noted that these studies excluded cysts classified as "difficult" or complicated cysts. Radical surgery was chosen based on the cyst's location. For instance, Ramia et al. [16] excluded cysts operated on in an emergency context. Others, such as Mohkam et al. [17] and Secchi et al. [18]. selected patients with cysts smaller than 10 cm. Yuksel et al. [19] and Motie et al. [20] excluded cysts complicated by large fistulas, cysts with intimate relationships with vessels, and intraparenchymal cysts. Moreover, a significant part of these comparative studies was conducted in specialized centers [21], indicating the presence of major selection biases in most of these studies. Due to these biases, the World Health Organization still advocates for conservative surgery as the method of choice in endemic areas [22].

Without doubt, radical treatment has shown promising results. but it may not be applicable in all cases, such as in cases of multiple cysts. Therefore, radical treatment and conservative treatment are not opposing techniques but rather complementary. Conservative surgery still plays an important role in endemic areas. It is crucial to try and improve its short and long-term outcomes through the standardization of surgical techniques.

The morbidity associated with conservative surgery for Hepatic Cystic Echinococcosis remains high despite many advances. Standardizing the conservative surgical technique could be the solution to reduce the rate of postoperative complications. It may also lead to a decrease in specific morbidity, which is the main factor to consider. Short-term complications of conservative treatment include residual cavity infection and external biliary fistula [19]. These

complications are the most frequent specific complications in conservative surgery, with a frequency ranging from 8.4% to 20.5% [8,23–26].

In the literature, several studies have focused on evaluating risk factors for Surgical Site Infections. These risk factors include communication with the bile ducts, thick pericyst and posterior cyst location.

Specific complications, especially DSSI, share a common denominator: bile leakage at the residual cavity postoperatively. Postoperative bile leakage is caused by cysto-biliary communication [19,23,27]. This allows us to regroup all these specific complications under the single term of DSSI since the share the same pathogenesis. DDSI represents a significant turning point in the management of Hepatic Cystic Echinococcosis. It transforms a simple and easily treatable parasitic cyst into a hepatobiliary pathology. Biliary fistula is a frequent complication, present in 37% of cases [28] and accounting for 60% of HCE complications [29].

Various predictive factors for occult cysto-biliary fistulas (CBF) have been identified in the literature. These include factors such as cyst size [30–32], thick pericyst [33], and elevated levels of liver enzymes and total bilirubin [34,35]. These risk factors are also often associated with a higher incidence of postoperative bile leak. Larger cysts often present with more extensive cyst walls and irregular internal surfaces, making it challenging for surgeons to detect and address all potential fistulas. Hence, there is a need for improvement and standardization in the management of HCE to effectively treat these CBF.

Biliary fistulas can be latent, occluded by the hydatid membrane, or patent. They can be identified during surgery or manifest postoperatively as an external biliary fistula or purulent retention [3,36]. Occult cysto-biliary fistulas are usually asymptomatic. The sensitivity of radiological examinations for detecting these fistulas varies widely among studies [29]. Frikha et al. [37] demonstrated the limitations of abdominal ultrasound and CT scan in detecting cysto-biliary fistulas. This makes their preoperative diagnosis challenging, highlighting the importance of intraoperative exploration.

Except in cases where the cyst content is stained with bile and there is a visible fistula at the bottom of the residual cavity, the intraoperative diagnosis remains difficult. The internal surface of the residual cavity is generally irregular and the pericyst is often thick, concealing small fistula orifices. Moreover, it is not always easy to explore the entire cavity (hepatic dome, intrahepatic cyst).

In 2011, Kayaalp et al. [8] conducted a comparative study aimed at studying the impact of intraoperative search for occult cysto-biliary fistulas on the DSSI rate. This study concluded that searching for and suturing cysto-biliary fistulas could reduce the postoperative bile leakage rate (8.8% vs. 27.7%; p = 0.03). which supports our findings. However, in this study, the search for fistulas was done with normal saline, which might miss small fistulas, possibly explaining the morbidity rate of 8.8%. Thus, neglecting silent occult cysto-biliary fistulas, especially the smallest ones (infra-millimetric) that can only be visualized through meticulous MBT testing, could be the cause of higher morbidity.

Methylene blue testing (MBT) appears to be the best examination for detecting occult cysto-biliary fistulas. In the literature, the technique of MBT was not well-defined (method of methylene blue injection, with or without clamping the main bile duct, leak test after suturing the fistula). MBT test in our study allowed us to detect a significant number of CBF in 53.7% of cysts.

Some authors postulate that the posterior liver segments are more prone to developing postoperative deep infections after conservative surgery [25,38,39]. This could be explained by several mechanisms such as the negative pressure exerted by the diaphragm, which would reverse the bile flow or having a suspended non-declining cavity [40]. For cysts located in the hepatic dome, the sectioning of the falciform, coronary and triangular ligaments allow for liver

**Table 4.  DSSI rates in the medical literature.**

|  | Study period | Country | Type | DSSI |
|---|---|---|---|---|
| **Monographie tunisienne [26]** | 2016–2021 | Tunisia | Observational Retrospective | 8.4% |
| **Al Saeedi et al. [24]** | - | Many | Meta-analysis | 14.7% |
| **Baraket et al. [23]** | 2001–2011 | Tunisia | Observational Retrospective | 16.6% |
| **Kayaalp et al. [8]** | 1975–2007 | Argentina | Observational Retrospective | 20.5% |
| **Our study** | 2017–2022 | Tunisia | Prospective | 1.9% |

mobilization to expose the cyst properly. This manoeuvre initially creates a dependent residual cavity, especially when involving the posterior liver segments, reducing the DSSI rates. Moreover, liver release better protects the peritoneal cavity with scolicidal-soaked gauzes. Thus, a cyst rupture could be easily controlled, preventing peritoneal cavity contamination by spillage, which is a source of recurrence. especially for posterior segments [41].

Several authors have demonstrated that a thick pericyst is a risk factor for the occurrence of deep surgical site infections [39]. Various technical measures have been employed to reduce complications related to the residual cavity. such as capitonnage, omentoplasty, or simple drainage. Among the studies that have addressed this question, only the study by Manterola et al. [42] showed that capitonnage resulted in less morbidity than omentoplasty. Two other randomized controlled trials [11,43] demonstrated the superiority of omentoplasty in reducing deep infections. The meta-analysis by Dziri et al. [9] showed a protective effect of omentoplasty compared to simple drainage of the cavity. Indeed, in our study, 67.2% of the cysts in the population underwent omentoplasty, contributing to reducing the DSSI rates.

As a result, conservative surgery still has a place in the therapeutic arsenal. The standardized technique, as we have described it, seems to yield better postoperative results. This standardization was achieved by considering various measures and techniques described in the literature. The absence of recurrence in the population may be attributed to the ameliorated techniques with systematic intraoperative ultrasound that is performed to detect all exogenous vesicles. Precautions taken, such as deflating the cysts before opening, during cyst puncture also contributes to decreasing recurrence.

The DSSI rate reported in the literature for conservative surgery, ranges from 8.4% to 20.5%, as presented in Table 4. The improvement in the specific morbidity rate in patients operated after the standardization of the technique is showed compared to the literature data.

The variability in the reported rates of complications could be attributed to the different definitions of external biliary fistula used by various authors. The criteria for defining a biliary fistula varied among studies, leading to differences in the interpretation and classification of postoperative bile leakages. Some authors considered any postoperative bile leakage through the drainage as a biliary fistula. Others defined a temporary bile leak as any bile leakage occurring within the first ten days after surgery and they considered it a biliary fistula if the bile leak persisted beyond the tenth day postoperatively.

The results of radical treatment (Table 5) reported in the literature show a low rate of overall morbidity ranging from 11% to 39%, as well as low rates of specific complications, ranging from 0% to 11% [14,18–20,26,44,45] and low rate of recurrence. However, these results were obtained at the cost of a longer operative duration. Comparing our results to those in the literature, we find that the rate of specific complications in the population treated with the standardized technique was at least as low as the radical treatment.

**Table 5. Results of radical management reported in literature.**

| Study | Study period | Country | Type | Overall morbidity | Specific morbidity | Recurrence |
|---|---|---|---|---|---|---|
| Akbulut et al. [46] | 2004–2009 | Turkey | Retrospective | 16.6% | 11.1% | 0 |
| Yuksel et al. [19] | 2001–2005 | Turkey | Prospective | 0 | 0 | 0 |
| Secchi et al. [18] | 1975–2007 | Argentina | Retrospective | 39% | - | 1.3% |
| Motie et al. [20] | 1993–2003 | India | Retrospective | 19% | 6.2% | 1.5% |
| Tagliacozzo et al. [47] | 1980–2005 | Italy | Retrospective | 16.2% | 4.6% | 1.2% |
| Baimakhanov et al. [5] | 2017–2019 | Kazakhstan | Prospective | 13.3% | - | 0 |
| Pang et al. [14] | 2016 | China | Meta-analysis | 18.4% | - | 2% |
| Farhat et al. [44] | 2000–2019 | Tunisia | Retrospective | 11.0% | 7.7% | 2.8% |
| Dziri et al. [13] | 2018 | Tunisia | Meta-analysis | 19.2% | 5.1% | 1.7% |
| Monographie tunisienne [26] | 2006–2021 | Tunisia | Retrospective | 14.9% | 11.2% | 1.0% |
| Our study | 2017–2022 | Tunisia | Prospective | 13.7% | 3.7% | 0 |

The standardized technique involves performing a cholecystectomy and cannulation of the cystic duct. These procedures are not without associated complications, such as bile duct injury or vascular injury during dissection. Moreover, performing intraoperative ultrasound requires specific skills and prior knowledge, such as understanding the functioning and settings of the ultrasound equipment. These factors, combined with the cost of the ultrasound device, may pose challenges to the widespread implementation of the standardized technique.

## Limits of our study

Our study has certain biases. The results of our study still remain hypothetical given the methodology adopted. To validate our conclusions, further comparative studies are necessary such as a prospective study comparing the standardised technique to the known partial cystectomy. Given the promising results of our study, a controlled randomized trial comparing radical surgery to the standardized technique could be conducted.

## Conclusion

Conservative surgery holds a significant place in the management of hepatic echinococcosis. However, the major drawback of conservative surgery is the high morbidity and recurrence rate. We studied a prospective cohort undergoing a standardized surgical technique for the conservative management and achieved results at least similar to radical management in terms of DSSI.

Several factors contributed to the success of the standardized technique in terms of DSSI. Liver mobilization by sectioning its attachments resulted in a more declivous residual cavity and improved exposure of posterior and upper segments. The use of methylene blue test before and after clamping detected occult CBF and allowed for adequate suturing, which could decrease specific complications. Finally, omentoplasty could reduce the DSSI rate. All these protective factors showed a statistically significant relationship with the occurrence of DSSI. By standardizing conservative surgery, it might be possible to achieve a specific morbidity rate similar to that of radical surgery while avoiding the morbidity associated with liver resection.

## Author Contributions

**Conceptualization:** Aymen Trigui, Rafik Mzali.

**Data curation:** Mohammad Saad Saumtally, Amira Akrout, Rahma Daoud, Wael Boujelbene.

**Formal analysis:** Aymen Trigui, Mohammad Saad Saumtally, Jihen Trabelsi.

**Investigation:** Aymen Trigui, Sami Fendri, Nozha Toumi, Salma Ketata.

**Methodology:** Aymen Trigui, Jihen Trabelsi.

**Project administration:** Aymen Trigui, Sami Fendri, Salah Boujelben.

**Resources:** Wael Boujelbene.

**Software:** Mohammad Saad Saumtally, Amira Akrout, Rahma Daoud, Wael Boujelbene.

**Supervision:** Rafik Mzali, Chadli Dziri, Salah Boujelben.

**Validation:** Chadli Dziri, Mohamed Ben Amar, Salah Boujelben.

**Writing – original draft:** Sami Fendri, Amira Akrout, Rahma Daoud, Nozha Toumi, Salma Ketata.

**Writing – review & editing:** Aymen Trigui, Chadli Dziri, Mohamed Ben Amar, Salah Boujelben.

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
