## [Decision Letter · Decision Letter 0]

28 Mar 2024

Dear Dr Saumtally,

Thank you very much for submitting your manuscript "Standardized approach to the conservative surgery of hepatic cystic echinococcosis: A prospective study" for consideration at PLOS Neglected Tropical Diseases. As with all papers reviewed by the journal, your manuscript was reviewed by members of the editorial board and by several independent reviewers. In light of the reviews (below this email), we would like to invite the resubmission of a significantly-revised version that takes into account the reviewers' comments. 

We cannot make any decision about publication until we have seen the revised manuscript and your response to the reviewers' comments. Your revised manuscript is also likely to be sent to reviewers for further evaluation.

Sincerely,

Husain Poonawala

Academic Editor

Uriel Koziol

Section Editor

In addition to carefully considering the reviewers´comments, please also include a comparison of your standardized conservative technique with previous publications of standardized conservative techniques (https://journals.plos.org/plosntds/article?id=10.1371/journal.pntd.0007516).

Reviewer's Responses to Questions

**Key Review Criteria Required for Acceptance?**

**Methods**

-Are the objectives of the study clearly articulated with a clear testable hypothesis stated?

-Is the study design appropriate to address the stated objectives?

-Is the population clearly described and appropriate for the hypothesis being tested?

-Is the sample size sufficient to ensure adequate power to address the hypothesis being tested?

-Were correct statistical analysis used to support conclusions?

-Are there concerns about ethical or regulatory requirements being met?

Reviewer #1: (No Response)

Reviewer #2: (No Response)

**Results**

-Does the analysis presented match the analysis plan?

-Are the results clearly and completely presented?

-Are the figures (Tables, Images) of sufficient quality for clarity?

Reviewer #1: (No Response)

Reviewer #2: (No Response)

**Conclusions**

-Are the conclusions supported by the data presented?

-Are the limitations of analysis clearly described?

-Do the authors discuss how these data can be helpful to advance our understanding of the topic under study?

-Is public health relevance addressed?

Reviewer #1: (No Response)

Reviewer #2: (No Response)

**Editorial and Data Presentation Modifications?**

Reviewer #1: (No Response)

Reviewer #2: (No Response)

**Summary and General Comments**

Reviewer #1: Thanks for authors for the efforts performed in this manuscript. I have many structural and scientific comments and I hope you accept:

1. Please get help of someone fluent in English. Many grammar errors are present. 

2. Abstract: 

1-patients:

• please add duration of study, multicenter study, stage of hydatid cyst and operative approach either open or laparoscope

• uncomplicated HCE then in exclusion criteria you used ----We excluded patients who had cysts complicated

2-results

• standardized techniques----please clarify

• 86%---no percentage without number of patients and vice verse. please strict to this information throughout the manuscript

• the cyst had a median size of 7.4 cm.---- please add minimum and maximum size

• 57 cysts-----please add percentage

• cystobiliary fistulas-----please differentiate between overt and occult hepatobiliary communication

• bile leak-----grade of biliary leak

3. Keywords: please add keywords

4. Introduction: 

• Hepatic Cystic Echinococcosis of the Liver (HCE)---- revise

• There are two main surgical approaches to treat HCE: conservative treatment and radical treatment ---add reference

• resection of the protruding dome (RPD) as described by Lagrot in 1957--- I think it is more suitable to use term partial cystectomy and omentoplasty. very old reference, please insert a more recent one

• As opposed to the latter technique, radical management

• which entails liver parenchymal resections with a non-negligible bleeding risk---please add reference

• references (3,4)---- reference 3 and 4 did not exclude complicated cysts

• Several technical refinements----references

5. patients and methods: 

• Please add a clear paragraph of inclusion and exclusion criteria. I think adding a flow chart of inclusion and exclusion criteria will be helpful.

• AF---please define

• Surgical procedure for the standardized technique--- please add a reference to the techniques used and then describe in short

• In case of CE1 and CE3a detected on intraoperative ultrasound----what about other stage of the cyst?

• he---- revise

• Injection of the dye is performed under low pressure without clamping of the common bile duct. These CBF are 157 sutured using Vicryl® 2/0. Clamping of the main bile duct while injecting the dye ---revise

• affixed to----how did you fixed it

• passive drainage of the residual cavity----site of drain insertion

• This study was conducted following the Helsinki Declaration—please add

• what is the protocol of albendazole therapy in your hospitals---please add

• please state if there are any cases lost to follow up

• please add statistical analysis paragraph

• please add a paragraph about Definitions of outcomes and measurements to include 

POBF was defined as a bilirubin concentration in the drain fluid at least three times higher than the serum bilirubin

concentration on or after postoperative day three[reference]. Hydatid recurrence was defined as the appearance of new active

cysts after surgery[reference]. Postoperative residual fluid cavities were not classified as recurrent HCD[reference]. Postoperative

morbidity was assessed using the Clavien-Dindo classification[reference]. Hydatid cyst fluid bilirubin and ALP levels were

measured using an automatic biochemical analyzer (AU-400, Olympus)[reference]. All hospitals used the same instrument to measure bilirubin and ALP levels in the cyst fluid. All the machines were calibrated at each institution to ensure consistent reporting. The time from surgery to disease relapse at any site was described as recurrence-free survival.

6. Results:

Table 1: 

• Recurrence--- what recurrence do you mean?

• Recurrence of PHE--- recurrence of pulmonary hydatid cyst?

• Cytolysis—clarify

• Hyperbilirubinemia--- please add bilirubin level

Table 2: 

o unique--- change to solitary

• WHO Classification--- change to please use WHO-IWGE classification

7. Discussion:

o need to be rewrite in a better way with starting paragraph about the aim of the study and important points in your study. Furthermore, please strict to your outcomes as I can see that the discussion is so long and many paragraphs are not needed as for example---In most hepatobiliary surgeries, the right subcostal approach was classically the most frequently performed. It allows for liver exposure and access to the elements of the hepatic pedicle. However, it may be less suitable for accessing the hepatic dome. The Makuuchi J-incision provides a better view of the posterior and 341 superior segments (27,28). This approach has the advantage of preserving the vascularization of the abdominal wall and causing fewer complications such as wall abscesses (29). This parameter seems to be correlated with liver mobilization, as it is the cysts located in the upper part of the liver (the dome) that require detachment. To access these cysts more easily, the Makuuchi incision is often preferred. 

o This could be explained by several mechanisms, such as the negative pressure exerted by the diaphragm, which would reverse the bile flow, or having a suspended non-declining cavity---please add a reference

Reviewer #2: (No Response)

PLOS authors have the option to publish the peer review history of their article (what does this mean?). If published, this will include your full peer review and any attached files.

Reviewer #1: Yes: Tamer.A.A.M.Habeeb

Reviewer #2: No
---

## [Decision Letter · Decision Letter 1]

12 Jun 2024

Dear Dr Saumtally,

We are pleased to inform you that your manuscript 'Standardized approach to the conservative surgery of hepatic cystic echinococcosis: A prospective study' has been provisionally accepted for publication in PLOS Neglected Tropical Diseases.

Best regards,

Husain Poonawala

Academic Editor

Uriel Koziol

Section Editor

Please see minor edit recommended by reviewer 2 - please address this during the proofreading stage.

Reviewer's Responses to Questions

**Key Review Criteria Required for Acceptance?**

**Methods**

-Are the objectives of the study clearly articulated with a clear testable hypothesis stated?

-Is the study design appropriate to address the stated objectives?

-Is the population clearly described and appropriate for the hypothesis being tested?

-Is the sample size sufficient to ensure adequate power to address the hypothesis being tested?

-Were correct statistical analysis used to support conclusions?

-Are there concerns about ethical or regulatory requirements being met?

Reviewer #1: (No Response)

Reviewer #2: (No Response)

**Results**

-Does the analysis presented match the analysis plan?

-Are the results clearly and completely presented?

-Are the figures (Tables, Images) of sufficient quality for clarity?

Reviewer #1: (No Response)

Reviewer #2: (No Response)

**Conclusions**

-Are the conclusions supported by the data presented?

-Are the limitations of analysis clearly described?

-Do the authors discuss how these data can be helpful to advance our understanding of the topic under study?

-Is public health relevance addressed?

Reviewer #1: (No Response)

Reviewer #2: (No Response)

**Editorial and Data Presentation Modifications?**

Reviewer #1: (No Response)

Reviewer #2: (No Response)

**Summary and General Comments**

Reviewer #1: (No Response)

Reviewer #2: I would like to congratulate the authors for their work. I have no further questions. Only in

abstract’s conclusion, authors should add “in selected patients” to ‘The standardized

conservative surgical technique shows promise in reducing DSSI rates and overall morbidity

and achieve as equally good result as radical management “. Otherwise I suggest the

acceptance for publication.

PLOS authors have the option to publish the peer review history of their article (what does this mean?). If published, this will include your full peer review and any attached files.

Reviewer #1: **Yes: **Tamer.A.A.M. Habeeb

Reviewer #2: **Yes: **Sepehr Abbasi Dezfouli

---

## [Editor Report · Acceptance letter]

18 Jun 2024

Dear Dr Saumtally,

We are delighted to inform you that your manuscript, "Standardized approach to the conservative surgery of hepatic cystic echinococcosis: A prospective study," has been formally accepted for publication in PLOS Neglected Tropical Diseases.

Best regards,

Shaden Kamhawi

co-Editor-in-Chief

Paul Brindley

co-Editor-in-Chief
